# CONSIDERATIONS FOR DISTRIBUTION SHIFT ROBUSTNESS IN HEALTH

**Arno Blaas**[*]**, Andrew C. Miller**[*]**, Luca Zappella, Jörn-Henrik Jacobsen, & Christina Heinze-Deml**
Apple
{ablaas,acmiller,lzappella,jhjacobsen,c_heinzedeml}@apple.com

## ABSTRACT

When analyzing robustness of predictive models under distribution shift, many works focus on tackling generalization in the presence of spurious correlations. In this case, one typically makes use of covariates or environment indicators to enforce independencies in learned models to guarantee generalization under various distribution shifts. In this work, we analyze a class of distribution shifts, where such independencies are not desirable, as there is a causal association between covariates and outcomes of interest. This case is common in the health space where covariates can be causally, as opposed to spuriously, related to outcomes of interest. We formalize this setting and relate it to common distribution shift settings from the literature. We theoretically show why standard supervised learning and invariant learning will not yield robust predictors in this case, while including the causal covariates into the prediction model can recover robustness. We demonstrate our theoretical findings in experiments on both synthetic and real data.

## 1 INTRODUCTION

In this work, we motivate how common assumptions in the domain generalization and invariant learning literature (Arjovsky et al., 2019; Ganin et al., 2016; Veitch et al., 2021) are violated in a set of broadly applicable problems, e.g. in healthcare. Invariant learning typically assumes a spurious or confounded association between outcome and covariates or auxiliary information. Hence, building a predictor that is invariant to the covariates or associated environment indicators can be shown to generalize better under distribution shift than standard empirical risk minimization. However, in many applications of interest for machine learning, e.g. in healthcare, there might not be only spurious associations between covariates and outcome, but also causal ones.[1]

One illustrative example is body mass index (BMI), which is causally related to a host of conditions, e.g., left ventricular hypertrophy (LVH) (Lorell & Carabello, 2000). BMI is not "spurious" in the sense that it is merely associated with LVH, but can directly cause changes in left ventricular mass, which in turn can lead to LVH (Himeno et al., 1996). However, a shift in the prevalence of elevated BMI can shift the association between a signal — e.g., an electrocardiogram (ECG) — that is influenced by both BMI and LVH.

We formalize such a causal setting and show that it leads to regression in performance of machine learning models under distribution shift, that can not be mitigated with common invariant learning methods. Our contributions are the following:

- We motivate and formalize a class of problems where covariates, such as demographics or other auxiliary data causally influence the outcome of interest and explain the difference to the commonly considered confounded or spurious associations.
- For this class of problems we show theoretically and on simulated data, how distribution shifts along such causally influencing covariates cause discrepancies in performance that can not be mitigated with invariant learning methods designed for the commonly considered confounded setting.

---

[*]These authors contributed equally to this work
[1]Unlike the fairness literature (Kilbertus et al., 2017; Kusner et al., 2017), we do not make a distinction as to whether this causal link is discriminatory or not.

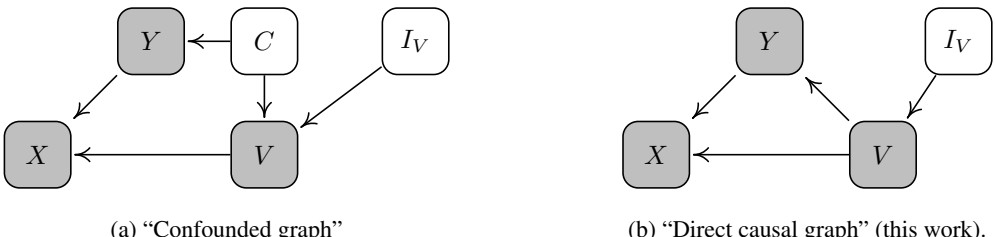

(a) "Confounded graph"   (b) "Direct causal graph" (this work).

Figure 1: Causal graphs considered in this work. (a) The "confounded graph" describes a spurious/confounded association between $Y$ and $V$, and has been considered in the ML literature (Heinze-Deml & Meinshausen, 2021; Veitch et al., 2021; Makar et al., 2022; Puli et al., 2022). This setting requires that the marginal $P(Y)$ remains invariant across distribution shifts. (b) In the "direct causal graph" (this work), the shortcut variable $V$ is a direct cause of the outcome $Y$, shifting the marginal $P(Y)$ when the intervention variable $I_V$ shifts the marginal $P(V)$.

## 2 THEORY

Consider predicting outcome $Y$ (e.g., health status) from features $X$ (e.g., an ECG recording) in the presence of an auxiliary covariate $V$ (e.g., age or BMI). One source of model brittleness can be "shortcuts", or features that are predictive in the training distribution, but not predictive under relevant distribution shifts (Arjovsky et al., 2019; Geirhos et al., 2020). To cope with such instability, one may try to remove the shortcuts during learning. One common approach to shortcut removal assumes a non-causal association between the potential shortcut $V$ and the outcome to be predicted $Y$ (Heinze-Deml & Meinshausen, 2021; Veitch et al., 2021; Makar et al., 2022; Puli et al., 2022), which, for instance, can arise due to a confounding covariate between $V$ and $Y$. The goal is then to seek a predictor using only $X$ that performs well across a range of distributions. For example, Makar et al. (2022) develop a risk invariant predictor across a family of related probability distributions motivated by the graph depicted in Figure 1a, that can be simplified for our analysis to

$$\mathcal{P}_{spur} = \{P_s(X \mid Y, V) \, P_s(Y) \, P_t(V \mid Y)\}, \qquad (1)$$

for a source distribution denoted by $s$ and shifted target distributions indexed by $t$. All target distributions in this family of distributions thus factor as $P_t(X, Y, V) = P_s(X \mid Y, V) \, P_s(Y) \, P_t(V \mid Y)$, i.e., they vary only in $P(V|Y)$ from the source distribution, while $P(X \mid Y, V)$ and $P(Y)$ remain unchanged. Notably, assuming that $P(Y)$ remains the same across all potential shifted distributions, can be an unrealistically strong assumption in applications like healthcare. For example, we would expect the prevalence of heart diseases ($Y$) to be higher in an older population ($V$).

Instead, in this work, we consider the scenario where the shortcut variable (e.g., age or BMI) is a direct causal parent of the outcome we wish to predict (e.g., myocardial infarction in an ECG), as depicted in Figure 1b. In this scenario, we wish to form good predictions for the family of distributions

$$\mathcal{P}_{cause} = \{P_s(X \mid Y, V)P_s(Y \mid V)P_t(V)\}. \qquad (2)$$

That is, we allow for changing marginal distribution of $P(V)$, while holding the conditional distributions $P(Y \mid V)$ and $P(X \mid Y, V)$ fixed.

Using the notion of so-called stable sets from Pfister et al. (2021), one can derive from the graph in Figure 1b which sets of predictors are associated with the same conditional expectation across different interventions on $V$ by checking which sets of covariates block all paths between $I_V$ and $Y$. Hence, in our model, to block the path $I_V \to V \to Y$, the covariate $V$ must be included in the set of predictors. The predictive distribution derived from the source that conditions on $X$ and $V$ is then invariant across the entire family, i.e., $P_s(Y \mid X, V) = P_t(Y \mid X, V)$, whereas the predictive distribution that only conditions on $X$ is not invariant, i.e., $P_s(Y \mid X) \neq P_t(Y \mid X)$ in general. We formalize this in the following proposition (proof in Appendix A.1).

**Proposition 1** *For any element $P_t \in \mathcal{P}_{cause}$ as defined in Eq. (2), it holds that $P_t(Y|X, V) = P_s(Y|X, V)$. Furthermore, for such a $P_t$, in general $P_t(Y|X) \neq P_s(Y|X)$.*

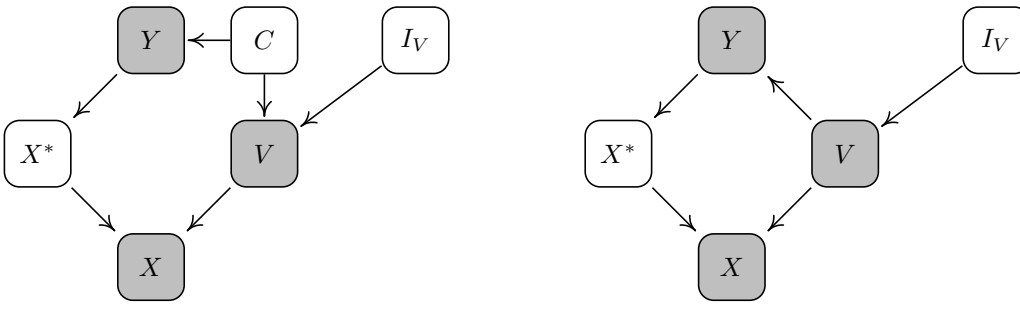

(a) "Extended confounded graph"  (b) "Extended direct causal graph" (this work).

Figure 2: Extended versions of causal graphs in Figure 1. (a) Graph considered in Makar et al. (2022) explicitly including invariant latent variable $X^*$ (still leading to shifts in $\mathcal{P}_{spur}$). Here, $X^*$ is a latent variable that describes variation in $X$ caused by $Y$. Recovering the predictive signal $e(X) = X^*$ yields a predictor that is invariant across interventions $I_V$, but one that does not use information from $V$ about $Y$. (b) Direct graph explicitly including $X^*$ (still leading to shifts in $\mathcal{P}_{cause}$). In this setting, recovering $X^*$ does not guarantee an invariant predictor across shifts due to $I_V$, demonstrated in Section 3

Hence, empirical risk minimization (ERM) using $\{V, X\}$ as predictors would yield a robust model with respect to $\mathcal{P}_{cause}$ while ERM using $\{X\}$ only would not.

**Remark 1** *Even an invariant representation that is invariant to $V$ and encodes only the information in $X$ related to $Y$ (e.g., $X^*$ in Makar et al. (2022)) would suffer from a degradation in performance across the family $\mathcal{P}_{cause}$.*

We illustrate these findings with a simulation study in the next section.

## 3 EXPERIMENTS

To illustrate our findings above regarding the consequences of shifts in causally influencing covariates $V$, we set up a simulation from a simple example.[2] To allow for analyses of the behaviour of invariant methods as well, we roll out the graphs from Figure 1 similar to Makar et al. (2022) by explicitly including an unobserved variable $X^*$ (see Figure 2). Here, $X^* = e(X)$ for some function $e$ is assumed to be a latent variable that only contains information about $X$ that is related to $Y$ and as such is invariant to $V$ when conditioned on $Y$.

In this extended setting (2), we define our data generating process as

$$p(V = 1) = p \tag{3}$$
$$P(Y = 1 \mid V = 0) = .2 \tag{4}$$
$$P(Y = 1 \mid V = 1) = .9 \tag{5}$$
$$P(X \mid Y = y, V = v) = \mathcal{N}(\mu_{y,v}, 1) \tag{6}$$
$$P(X^* \mid Y = y) = \mathcal{N}(\mu_{y,0}, 1). \tag{7}$$

where $\mu_{0,0} = -2/3, \mu_{1,0} = 2/3, \mu_{0,1} = -.8$, and $\mu_{1,1,} = .8$. As such, it is constructed to allow to analyse shifts of the family $\mathcal{P}_{cause}$, where $P(V)$ can be shifted by varying $p$ while $P(Y|V)$ and $P(X|Y,V)$ will remain the same.

In Figure 3, we compare the performance of the predictors $P_s(Y \mid X)$, $P_s(Y \mid X^*)$ and $P_s(Y \mid X, V)$ on distributions where the marginal $P(V)$ has been shifted, and the source distribution marginal is $P_s(V = 1) = .1$. Predictors are obtained in closed form, and performance metrics are calculated on a sample of size 20,000 drawn according to the source distribution. As discussed in Section 2, this shift induces a shift in $P(Y)$, $P(Y \mid X)$, *and* $P(Y \mid X^*)$, causing a

---

[2]First results of experiments on real data (annotated electrocardiogram or ECG recordings in the PTB-XL data set (Wagner et al., 2020)) can be found Appendix A.2.

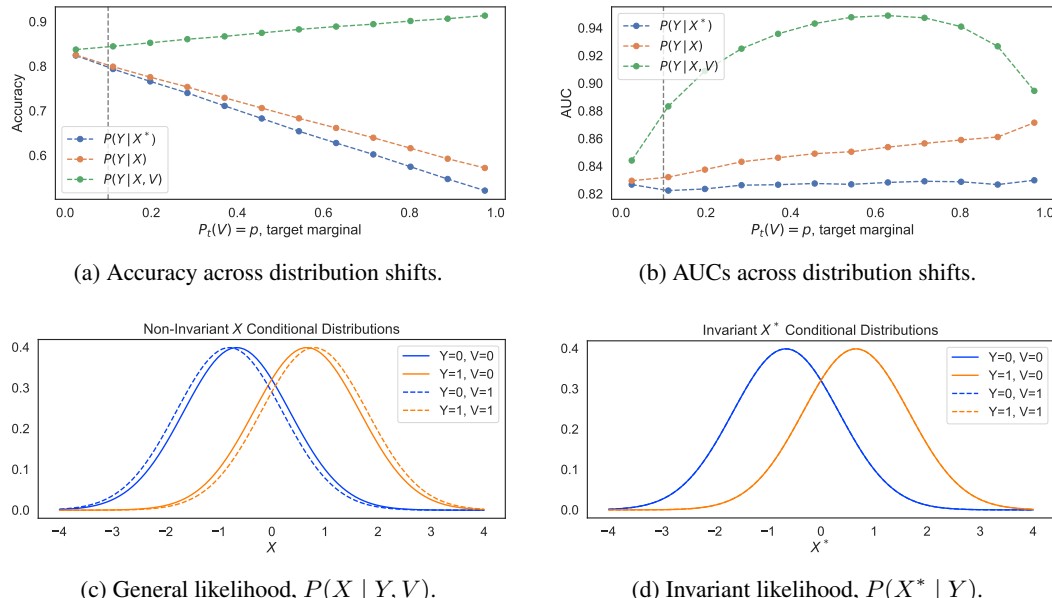

(a) Accuracy across distribution shifts.

(b) AUCs across distribution shifts.

(c) General likelihood, $P(X \mid Y, V)$.

(d) Invariant likelihood, $P(X^* \mid Y)$.

Figure 3: Simulation study described in Section 3. Panel (a) compares the predictive accuracy of the models $P_s(Y \mid X)$, $P_s(Y \mid X^*)$ and $P_s(Y \mid X, V)$ as a function of the target marginal $P_t(V) = p$, with source $P_s(V) = .1$. Only $P_s(Y \mid X, V)$ does not degrade in accuracy as $P_t(V)$ shifts further away from $P_s(V)$. Panel (b) compares the predictive AUC of the same three models. Note that the predictor using $X^*$, $P_s(Y \mid X^*)$ does achieve invariance in AUC across shifts, but not accuracy (or likelihood), and cannot make use of information about $Y$ from $V$. Panel (c) depicts the four likelihood models (one for each combination of $Y$ and $V$) — note that $V = 1$ further separates the conditional distributions, making separation easier (hence the AUC goes up in Panel (b) as $p$ increases). Panel (d) depicts a $P(X^*|Y, V)$, which is the same across values of $V$ (unlike $P(X|Y, V)$ in Panel (c)). The overall key takeaway is the robustness of $P_s(Y \mid X, V)$, i.e. the model conditioning on both $V$ and $X$ versus the lack of robustness in models conditioning only on $X$ or $X^*$ in terms of predictive accuracy in Panel (a) (even when their AUC is robust across shifts, Panel (b)).

degradation in performance of the predictors $P_s(Y \mid X)$ and $P_s(Y \mid X^*)$, but not of $P_s(Y \mid X, V)$, i.e. conditioning on $V$ restores performance across distribution shifts.

As a side note, note that the AUC performance of $P(Y \mid X^*)$ does not degrade, though a general risk (like accuracy or log-likelihood) does degrade. This is due to the fact that shifts in $P(V)$ only influence $P(Y \mid X^*)$ through the prevalence $P_t(Y) = \sum_{v'} P_s(Y \mid V = v')P_t(V = v')$, not through $X^*$. The AUC metric is invariant to prevalence, but general metrics like accuracy, log-likelihood, and calibration are sensitive to prevalence. Also note that the difference in AUC performance of $P(Y \mid X^*)$ and $P(Y \mid X)$ is due to the construction of the class conditional distributions depicted in Figures 3c and 3d.

Overall, the degradation (or robustness) of performance across shifts of the family $\mathcal{P}_{cause}$ is the main illustrative point to be observed in Figure 3 and this section.

## 4 DISCUSSION

Our theoretical findings show that for settings in which auxiliary covariates $V$ causally influence the outcome of interest $Y$ (rather than just being spuriously correlated to them), $P(Y|X, V)$ remains stable across shifts in $P(V)$, while $P(Y|X)$ in general does not. As such, regressing $Y$ only on $X$ to learn $P(Y|X)$ (or invariant derivations thereof) will lead to predictions that are not robust to such shifts, while regressing $Y$ on $X, V$ recovers the desired robustness, as we empirically demonstrate on simulated data. Lastly, we also demonstrate the former (regressing $Y$ on $X$ not being robust to such shifts) on a real world healthcare application that consists in predicting ECG statements based on ECG recordings. We plan to extend this experiment to also demonstrate the latter.

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

## A   APPENDIX

### A.1   PROOF OF PROPOSITION 1

First, we show that for any element $P_t$ of this family, it holds that $P_t(Y|X,V) = P_s(Y|X,V)$. Remember that by the definition of $\mathcal{P}_{cause}$, we have that $P_t(X \mid Y,V) = P_s(X \mid Y,V)$ and $P_t(Y \mid V) = P_s(Y \mid V)$. From this, it quickly follows that

$$P_t(X|V) = \int P_t(X|Y,V)P_t(Y|V)dY \tag{8}$$

$$= \int P_s(X|Y,V)P_s(Y|V)dY \tag{9}$$

$$= P_s(X|V) \tag{10}$$

Then, using basic probability calculus, it follows

$$P_t(Y|X,V) = \frac{P_t(Y,X,V)}{P_t(X,V)} \tag{11}$$

$$= \frac{P_s(X|Y,V)P_t(Y|V)P_t(V)}{P_t(X|V)P_t(V)} \tag{12}$$

$$= \frac{P_s(X|Y,V)P_s(Y|V)}{P_t(X|V)} \tag{13}$$

$$= \frac{P_s(X|Y,V)P_s(Y|V)}{P_s(X|V)} \tag{14}$$

$$= \frac{P_s(X|Y,V)P_s(Y|V)P_s(V)}{P_s(X|V)P_s(V)} \tag{15}$$

$$= P_s(Y|X,V) \tag{16}$$

Next, we show that for such an element $P_t$ of this family, in general $P_t(Y|X) \neq P_s(Y|X)$. Using the above result, this is indeed easy to see when marginalising over $V$:

$$P_t(Y|X) = \int P_t(Y|X,V)P_t(V|X)dV \tag{17}$$

$$= \int P_s(Y|X,V)P_t(V|X)dV \tag{18}$$

$$= \int P_s(Y|X,V)\frac{P_t(X|V)P_t(V)}{P_t(X)}dV \tag{19}$$

$$= \int P_s(Y|X,V)P_s(X|V)\frac{P_t(V)}{P_t(X)}dV \tag{20}$$

$$\tag{21}$$

Since in general $\frac{P_t(V)}{P_t(X)} \neq \frac{P_s(V)}{P_s(X)}$, this also implies that in general $P_t(Y|X) \neq P_s(Y|X)$.

### A.2   EXPERIMENTS ON REAL DATA

#### A.2.1   EXPERIMENTAL SETUP

Here, we include some first results of experiments on real data, which demonstrate the lack of robustness of models trained only on $X$ (i.e. trained to learn $P(Y|X)$) to shifts of the family $\mathcal{P}_{cause}$, as predicted by our theory and synthetic experiments in the main body. Extending these experiments and showing that training on $X, V$ can achieve higher robustness is left for future work.

**Data**   We base our experiments on the PTB-XL data set (Wagner et al., 2020). It contains 21,837 clinical 12-lead ECG recordings from 18,885 patients. Each recording is 10 seconds long and is processed following previous literature at a frequency of 100 Hz (Strodthoff et al., 2021).

Each ECG recording is annotated with ECG statements that can be grouped into 5 classes of superdiagnostics: normal ECG (NORM), conduction disturbance (CD), myocardial infarction (MI), hypertrophy (HYP), and ST/T changes (STTC). We base our analysis on the 4 superdiagnostics that indicate abnormal ECG.

Furthermore, the data set contains an annotation of age for 21,748 of the recordings. These will be the annotations we use as $V$-variable. To this end, we discretise age to get $V_A \in \{< 50, \ 50 - 63, \ 63 - 74, \ 74+\}$ as demographic variable indicating age bin. These age thresholds were chosen to yield sufficient samples per bin, and as such approximately correspond to the quartiles.

**Model and training**   We base our analysis on the best performing model in the supervised deep learning benchmark analysis by Strodthoff et al. (2021), a xresnet1d101 model. We use the publically available tsai library (Oguiza, 2022) to implement our xresnet1d101 model and obtain clean test set performance that is similar to the one reported in Strodthoff et al. (2021) (using the same training, validation, and test splits).

**Introducing distribution shifts**   Next, we introduce some shifts in the test set by changing $P(V)$ in comparison to the original test set (which follows the same distribution as the training set). To this end, we sample subsets from the test set that are constrained to be $90\%$ older than a certain age threshold, increasing that age threshold in steps of 10 years from 40 to 80. E.g., the shifted evaluation set '90% 70+' consists of $90\%$ of samples from the test set from people that are over 70, and $10\%$ from people that are at most 70 years old. We always use the maximal possible subset size, which diminishes as we increase the threshold (as in the original test set there are, e.g., more $40+$ than $80+$ samples), making the evaluation set sizes for the age shifts $n_A = 2042, 1765, 1312, 775, 306$ from '90% 40+' to '90% 80+'. We take the randomness introduced through this subsampling into account by showing average results and standard deviations over 5 runs.

### A.2.2   VERIFYING THAT SHIFTS BELONG TO $\mathcal{P}_{cause}$

First, we verify that the (target) distributions resulting from the shifts we introduce above belong to the family $\mathcal{P}_{cause}$, i.e. that while $P(V)$ changes, $P(Y|V)$ and $P(X|Y,V)$ remain unchanged. We can verify that $P(Y|V)$ remains unchanged by counting. For age, in Figure 4, we verify that indeed $P_t(Y|V_A) \approx P_s(Y|V_A)$ for all $t$ under inspection. Here, the source distribution $s$ is the one of the original test set (column 'Original'), and we can see that for all the target distributions $t$ introduced by the shifts in A.2.1 (remaining columns), $P(Y|V_A = v)$ remains roughly constant (inside standard deviations) for all $v \in \{< 50, \ 50 - 63, \ 63 - 74, \ 74+\}$. Also, the difference between $P(Y|V_A = v_i)$ and $P(Y|V_A = v_j)$ remains prominent for all $v_i \neq v_j \in \{< 50, \ 50 - 63, \ 63 - 74, \ 74+\}$ for all superdiagnostics (and almost all shifts). We would ideally also check that $P(X|Y,V)$ remains unchanged across shifts. Since $X$ is not binary but rather continuous and high-dimensional, this would require statistical tests such as presented in Gretton et al. (2012), something that we leave for future work (note that both $\mathcal{P}_{spur}$ and $\mathcal{P}_{cause}$ assume $P(X|Y,V)$ to remain unchanged). One additional thing to note is that for the shifts we consider, $P(Y)$ changes, i.e. they are definitely not part of $\mathcal{P}_{spur}$ (see Figure 5).

### A.2.3   REGRESSING $Y$ ONLY ON $X$ IS NOT ROBUST UNDER SHIFTS IN $P(V)$

In Figure 6, we can observe what happens to the predictive accuracy (Acc) and the area under the receiver operating characteristic curve (AUC) under the shifts of the family $\mathcal{P}_{cause}$ introduced above when the predictive model is regressing $Y$ on $X$. For the shifts along age, for all 4 superdiagnostics (CD, HYP, MI, STTC), regressing $Y$ only on $X$ is not robust under shifts of the family $\mathcal{P}_{cause}$ in terms of neither accuracy nor AUC, as predicted by the theory in Section 2. For all superdiagnostics, we observe a monotonic drop of accuracy and AUC as the evaluation set gets shifted to an increasingly older subpopulation of the test set, reaching around $5\%$ drop in accuracy for the evaluation set consisting of $90\%$ of people older than 80.

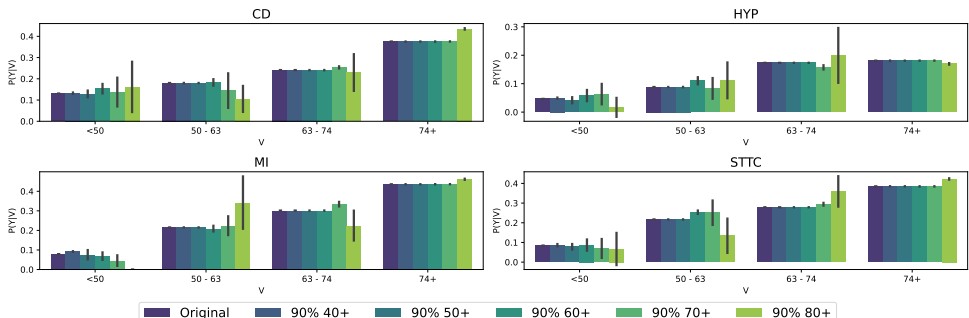

Figure 4: Validation of shifts for V=Age. For each of the 4 superdiagnostics (subplots), we show $P(Y|V_A = v)$ per $v \in \{< 50, \ 50 - 63, \ 63 - 74, \ 74+\}$ (leftmost group of bars in subplot to rightmost group of bars in subplot) for the original, unshifted test set (dark purple) and the shifts introduced in Section A.2.1. Equal height of bars inside each group of bars is indicative for constant $P(Y|V)$ across evaluation sets.

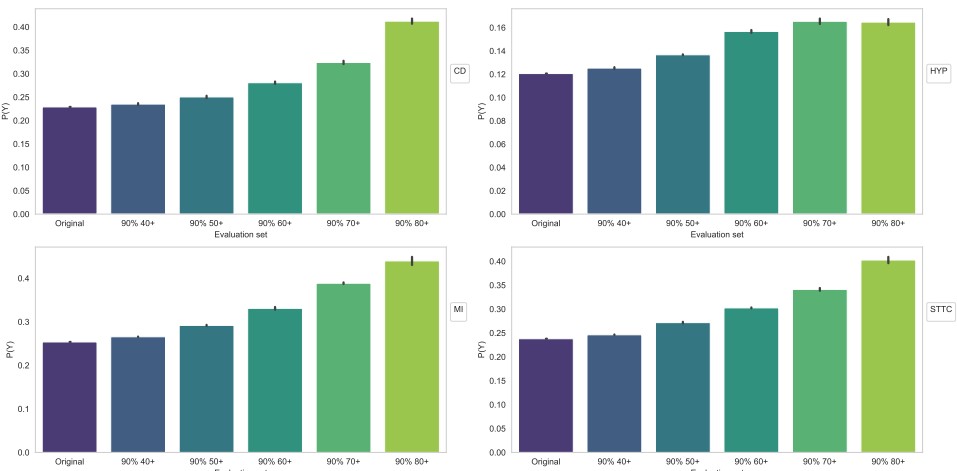

Figure 5: Changes of $P(Y)$ induced by shifts along V=Age. For each of the 4 superdiagnostics we show $P(Y)$ for the original, unshifted test set (dark purple) and the shifts introduced in Section A.2.1. Clearly, it does not remain unchanged, thus confirming that the shifts do not belong to the family $\mathcal{P}_{spur}$.

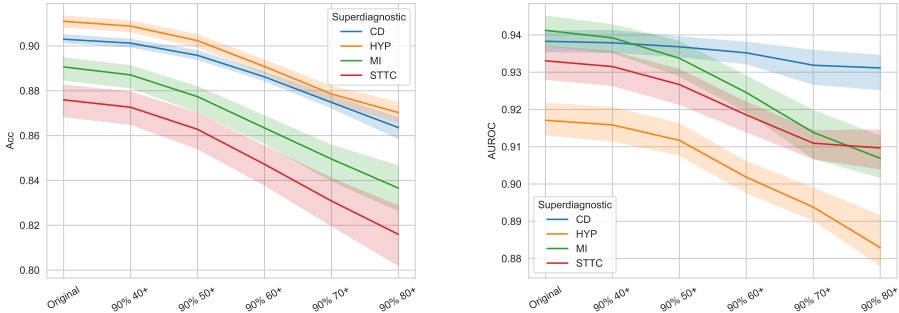

Figure 6: Accuracy (left) and AUC (right) for age distribution shifts for xresnet1d101 model trained on original training data when evaluated on shifted evaluation sets. Only the original test set ('Original') follows the same distribution as the training data, all others follow shifted distributions introduced in Section A.2.1 that belong to the family $\mathcal{P}_{cause}$, as verified in Section A.2.2. Average across 5 runs, with shaded standard deviations.

