# OpenReview forum: "Considerations for Distribution Shift Robustness in Health"
_ICLR.cc/2023/Workshop/TML4H — ICLR 2023 Workshop TML4H Oral_

### Official Review · Reviewer_mQEu · 2023-03-01
**Interesting work on the effectiveness of invariant learning under causal association**

**Rating:** 6
**Confidence:** 3

**Review:**

This paper formalizes and studies predictive problems where covariates (e.g. age) are *causally* related to the target outcome (e.g. heart diseases), and shows that existing techniques for removing *spurious* correlations cannot improve robustness to such distribution shifts. The authors present a theoretical analysis of causal graphs for cases when covariates and targets are spuriously or causally associated, showing that distributional robustness can only be achieved by conditioning the prediction on the *causal* covariates. This is validated in simulation study, where predictor conditioned on the covariate shows the best performance under distribution shifts.

Pros:
- The paper is overall well written and easy to follow.
- The paper presents an interesting perspective on the well studied problem of domain generalization, identifying key limitations in the presumption of existing approaches.
- The theoretical analysis is concise yet convincing. The simulated experiment clearly reveals the failure case of simple invariant learning under the causal setting.

Cons:
- No empirical results on real data. It would be interesting to study how the causal links between targets and covariates (e.g. heart disease and age) compromise the performance of prior invariant learning methods.
- More complex simulated scenarios would be more helpful, such as a hybrid between graph (a) and (b) in the paper, where target Y is determined by both V and another confounding variable C. It would also make sense to use non-binary or continuous values of Y and V.
- There can be more discussion to related works in the fair machine learning literature, e.g. causal fairness [a,b], which also seek to achieve invariance under causal interventions.

[a] Avoiding Discrimination through Causal Reasoning, Kilbertus et al.

[b] Counterfactual Fairness, Kusner et al.

---

### Official Review · Reviewer_Giej · 2023-03-01
**Comments to TML4H, paper26**

**Rating:** 6
**Confidence:** 4

**Review:**

The authors of this study examine a type of distribution shift that involves covariates that are causally associated with outcomes of interest, rather than being spurious. This scenario is often encountered in the health field, where causal relationships between covariates and outcomes are common. The authors formalize this situation and compare it to other distribution shift scenarios discussed in previous literature. They demonstrate why standard supervised learning and invariant learning methods are inadequate for producing robust predictions in this case, but incorporating causal covariates into the prediction model can restore robustness. The authors validate their theoretical results by conducting experiments on synthetic data.

(1) This paper is difficult to follow, and there are some awkward expressions that make it harder to understand.

(2) The problem under study holds great importance in the field of health research.

(3)The new idea is supported by both experimental evidence and theoretical analysis.

(4) Although the paper's title is "Considerations for Distribution Shift Robustness in Health," the proposed method has not been validated on actual health data. Therefore, I suggest that the proposed method be evaluated on real health data and compared to state-of-the-art models to determine its efficacy.

---

### Meta-Review · Area_Chair_CScR · 2023-03-05

**Recommendation:** Accept (Poster)
**Confidence:** 5

**Metareview:**

The paper studies an important problem where covariates are causally associated with the studied diseases. With theoretic analysis and experiments on simulated data, the authors demonstrated that neither supervised learning nor invariant learning can address such a condition, which poses a new direction of challenge to the community. I also agree with the reviews that experiments on more realistic scenarios would be much more helpful to further validate the finding and draw more attention from the community.